



# Drainage assessment of irrigation districts: on the precision and accuracy of four parsimonious models

Pierre Laluet[1], Luis Olivera-Guerra[1,*], Víctor Altés[2,3], Vincent Rivalland[1], Alexis Jeantet[4], Julien Tournebize[4], Omar Cenobio-Cruz[5], Anaïs Barella-Ortiz[5], Pere Quintana-Seguí[5], Josep Maria Villar[3], Olivier Merlin[1]

[1]Centre d'Etudes Spatiales de la Biosphère (CESBIO), Université de Toulouse, CNES, CNRS, IRD, UPS, Toulouse, France
[2]isardSAT, Marie Curie 8-14, Parc Tecnològic Barcelona Activa, Barcelona, Spain
[3]Department of Environment and Soil Sciences, Universitat de Lleida, Lleida, Spain
[4]University of Paris-Saclay, INRAE Jouy-en-Josas – Antony, UR HYCAR, Antony, France
[5]Observatori de l'Ebre, Universitat Ramon Llull – CSIC, Roquetes, Spain
*now at: Laboratoire des Sciences du Climat et de l'Environnement, CEA-CNRS-UVSQ-UPSACLAY, UMR 8212, IPSL, Gif-sur-Yvette, France

Correspondence: pierre.laluet@gmail.com

**Abstract.** In semi-arid irrigated environments, the agricultural drainage is at the heart of three agro-environmental issues: it is an indicator of water productivity, it is the main control to prevent soil salinization and waterlogging problems, and it is related to the health of downstream ecosystems. Crop water balance models combined with subsurface models can be used to estimate the drainage quantities and dynamics at various spatial scales. However, the precision (capacity of a model to fit the observed drainage using site-specific calibration) and accuracy (capacity of a model to approximate observed drainage using default input parameters) of such models have not yet been assessed in irrigated areas. To fill the gap, this study evaluates four parsimonious drainage models based on the combination of two surface models (RU and SAMIR) and two subsurface models (Reservoir and SIDRA) with varying complexity levels: RU-Reservoir, RU-SIDRA, SAMIR-Reservoir, and SAMIR-SIDRA. All models were applied over two sub-basins of the Algerri-Balaguer irrigation district, northeastern Spain, that are equipped with surface and subsurface drains driving the drained water to general outlets where the discharge is continuously monitored. Results show that RU-Reservoir is the most precise (average KGE ($Q^{0.5}$) of 0.87), followed by SAMIR-Reservoir (average KGE ($Q^{0.5}$) of 0.79). However, SAMIR-Reservoir is the most accurate model for providing rough drainage estimates using the default input parameters provided in the literature.



## 1 Introduction

In the context of ongoing global changes, semi-arid irrigated areas especially face multiple challenges. First, the agricultural water productivity is a critical issue in regions where water resources are under increasing pressure (FAO, 2021). Second, one-third of the world's irrigated land is affected by the soil salinization issue, which is likely to bring a significant loss of arable lands (Singh et al., 2019). Third, the non-point pollution is another issue

over irrigated areas, with return flows that may contain high concentrations of nutrients (García-Garizábal et al., 2012) and/or pesticides (Abdi et al., 2021).

At the heart of the above three challenges (water productivity, soil salinization and non-point pollution) lies the agricultural drainage. More than 20% of the total irrigated lands in the world is equipped with drainage systems including open ditches or buried drains (Schultz et al., 2007). Drainage systems are generally installed with the

aim to prevent waterlogging during heavy rainfall events (by the sudden rise of groundwater levels) and to facilitate salt leaching. Moreover, the drained water quantity and quality are strong indicators of the agricultural water productivity, and of the possible impact of nitrates, salts, and pesticides concentration on downstream ecosystems (Blann et al., 2009).

In this context, being able to estimate the drained water in irrigated areas is of major importance, whether by

monitoring the quantity and quality of the drained water discharged (e.g., Negm et al., 2017), or by developing drainage scenarios that integrate changes in agricultural practices (e.g., Tournebize et al., 2004) or in climatic conditions (e.g., Golmohammadi et al., 2020; Jeantet et al., 2022). To date, only a few studies have dealt with the quantitative estimation of drainage in semi-arid irrigated areas. At the field scale, Ale et al. (2013) compared the ability of the physically-based DRAINMOD (Skaggs et al., 2012) and ADAPT (Gowda et al., 2012) models to

simulate monthly drainage in a drip-irrigated plot in the US. The determination coefficient between simulated and observed drainage was 0.90 and 0.85 for ADAPT and DRAINMOD, respectively, for data during a seven-year period. More recently, Feng et al. (2021) simulated the daily drainage with the physically-based Hydrus-2D model in a furrow-irrigated plot and obtained a Nash-Sutcliffe model efficiency coefficient (NSE) value of 0.91 and 0.94 for the calibration and validation year respectively. At a larger spatial scale, Cavero et al. (2012) simulated the

monthly drainage of three Mediterranean irrigated catchments (mostly surface irrigation) located in Spain, Algeria, and Turkey, and ranging from 4000 to 10000 hectares. They used the crop water balance model APEX (Gassman et al., 2009) coupled with DRAINMOD over two hydrological years and obtained Root-mean-square deviation (RMSD) ranging from 3.4 mm per month to 25.3 mm per month. At much larger scale, Wen et al. (2020) simulated the monthly drainage over 19 sub-basins in a 1.2 million hectare irrigation district in northern China using an

empirical approach based on water table observations. The results showed a mixed performance with an average NSE of 0.64 and a standard deviation of 0.21 for the 2-year calibration period, and an average NSE of 0.34 and a standard deviation of 0.44 for the 2-year validation period. In the same irrigation district, Chang et al. (2021) simulated the annual drainage discharge under different management scenarios using the semi-empirical SAHYSMOD model (Oosterbaan et al., 2005).

Although the literature on the agricultural drainage is still limited for semi-arid irrigated areas, there are many scientific papers on the drainage estimation in humid areas, where waterlogging problems are common. The vast



majority of them use physically-based models such as DRAINMOD (Moursi et al., 2022; Muma et al, 2015), RZWQM2 (Ma et al., 2012; Xian et al., 2017; Jiang et al. 2020), MACRO (Larsbo et al., 2005; Jarvis and Larsbo, 2012), SWAP (van Dam et al., 2008), HydroGeoSphere (De Schepper et al., 2015), and FLUSH (Turunen et al.,
2013; Nousiainen et al., 2015). Generally, these models are implemented at the plot scale and represent the intermediate processes (e.g., macropore infiltration, deep seepage, water redistribution in the soil profile, rooting distribution, lateral flows) involved in drainage at the daily or hourly time step. They rely on a lot of information for model parameterization, implying a detailed knowledge of the studied site, and potentially numerous parameters to calibrate. For example, Ma et al. (2012) recommend for the RZWQM2 model an independent
measurement for 11 parameters, and a calibration for 11 others, out of a total of 24 parameters (the two remaining ones being taken from the literature). In fields where intensive measurement campaigns have been conducted, these models can simulate well the observed drainage at hourly, daily, weekly, or monthly scales. However, due to the need for site-specific calibration (using drainage measurements) to set their relatively numerous input parameters, the application of such models to poorly monitored basins remains limited.

With the aim to generalize a drainage model for various agricultural conditions, Henine et al. (2022) proposed a simple semi-empirical drainage model, RU-SIDRA. It combines a surface model (RU) to simulate the daily recharge and a subsurface model (SIDRA) (Lesaffre and Zimmer, 1987; Bouarfa and Zimmer, 2000) to convert the simulated recharge into daily drainage discharge. RU is a water balance model based on a simplified version of the FAO-56 method and relies on a single sensitive parameter only. SIDRA is based on the resolution of a semi-
analytical formula derived from the Boussinesq physical equation (Boussinesq, 1904), leading to two main sensitive parameters. The robustness of RU-SIDRA was evaluated by Jeantet et al. (2021) on 22 non-irrigated French fields and sub-basins over 200 hydrological years. It was found that RU-SIDRA performs as well as physically-based models in reproducing daily drainage and is as robust as the latter from one hydrological year to another. However, Jeantet et al. (2021) emphasized a limitation of RU-SIDRA associated with the empirical nature
of the RU model. Indeed, in RU, the start of drainage is not systematically well reproduced from year to year. This is potentially due to its poor representation of the processes governing the variation of the soil water stock (e.g., root growth or evapotranspiration). Over irrigated areas, this difficulty is expected to be further exacerbated by the impact of summer crops. In addition, as the physically based-models described previously, RU-SIDRA also relies on a site-specific calibration step using rarely available observed drainage data.

Finding the right balance between the simplicity required for a drainage model to be easily applicable over large areas and the complexity required to ensure a sufficient realism of intermediate processes and hence the robustness of drainage estimates in time remains a challenge. In this context, this study seeks to address the following questions:

- Can parsimonious models with different degrees of complexity precisely reproduce the daily drainage in a semi-
arid irrigated context with site-specific calibration?

- Can such models with default calibration (with parameter values provided in the literature) reproduce drainage quantities and dynamics, even roughly?



In this context, we evaluated the precision and accuracy of several parsimonious models based on the RU-SIDRA formalism. By precision and accuracy, it is meant the capacity of each model to predict the drainage after site-specific calibration (using drainage measurements) and by setting the model input parameters to default values found in the literature (without using drainage measurements), respectively.

To cover a range of modeling complexity, we investigate the SAMIR model (Simonneaux et al., 2009) as an alternative to the RU model. SAMIR is more complex than RU as it simulates more processes (e.g., root growth, vegetation development, evaporation, vegetation cover, specific crop water needs and resistance to stress), while remaining parsimonious with only two parameters integrating most of the sensitivity for the recharge simulation (Laluet et al., 2023). We also investigate the Reservoir model as an alternative to the SIDRA model. Reservoir is a fully empirical model driven by a single parameter. It is a simplified version of a module recently incorporated in the SASER (Safran-Surfex-Eaudysee-Rapid) hydrological model (Quintana-Seguí et al., 2017; Vergnes and Habets, 2018; David et al., 2011) by Cenobio-Cruz et al. (2023) who showed its ability to satisfactorily reproduce the low flows (groundwater discharge) observed at 53 hydrological stations in France and Spain.

The combination of both recharge (RU and SAMIR) models and both subsurface (Reservoir and SIDRA) models generates four drainage models by order of increasing complexity: RU-Reservoir (two main parameters), RU-SIDRA (three main parameters), SAMIR-Reservoir (three main parameters), SAMIR-SIDRA (four main parameters). In this study, the precision and accuracy evaluation of the four drainage models are carried out in two sub-basins of the Algerri-Balaguer irrigation district located in the Ebro basin, northeast of Spain. These two sub-basins are both instrumented with flow meters continuously measuring the daily drainage discharge into the main drains.



## 2 Material and methods

The overall methodology to assess both precision and accuracy of RU-Reservoir, RU-SIDRA, SAMIR-Reservoir, and SAMIR-SIDRA, is presented in the flowchart of Fig. 1. First, the study site and data used are presented (Section 2.1), followed by a description of the four models (Section 2.2). The following sections describe the site-specific calibration strategy used for evaluating the precision of the models (Section 2.3) and the selection of input parameter ranges used for evaluating the accuracy of the models (Section 2.4).

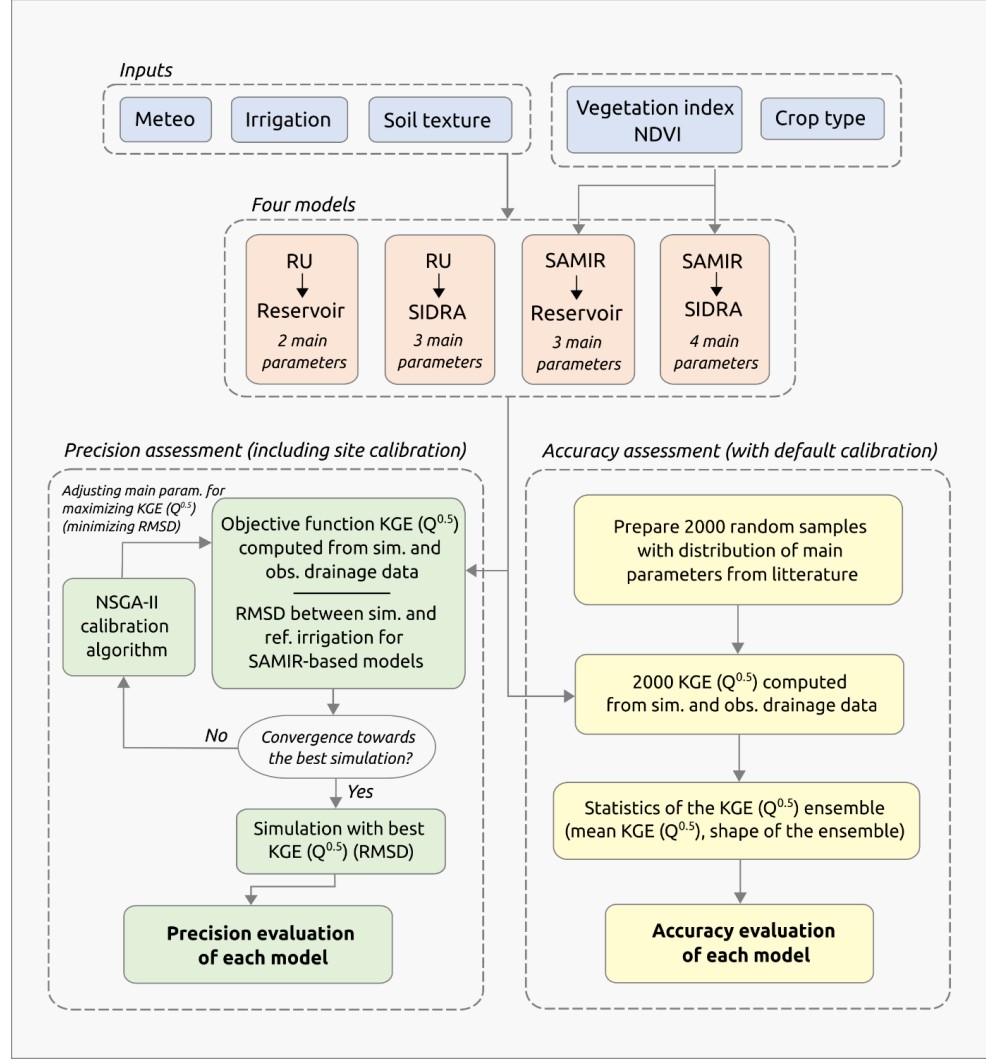


**Figure 1.** Flowchart of the proposed methodology for precision and accuracy evaluation of the drainage simulated by four parsimonious models with different levels of complexity.



### 2.1 Study area and data

**2.1.1 Study area**

The Algerri-Balaguer (AB) irrigation district is located in northeast Spain, 20 km north of the city of Lleida. It is characterized by a semi-arid continental Mediterranean climate with an average annual reference evapotranspiration ($ET_0$) of 1027 mm and precipitation of 380 mm (2000-2021). AB has an area of 8100 hectares of cropland with mainly corn, barley, wheat, fruit trees, and alfalfa. 6800 hectares are equipped for irrigation with

sprinklers for annual crops and drip systems for fruit trees. An overview of the AB area is shown in Fig. 2. For an extensive description of the irrigation district in terms of soil, geology, crops, irrigation and drainage system the reader is referred to refer to Altés et al. (2022).

A network of surface (open ditches) and subsurface (buried pipes) drains was installed in the AB district from 1998 to 2017, intercepting a water table relying on a shallow impermeable layer (Altés et al., 2022). Field drains

(underground perforated plastic pipes) are connected to main drains that convey the water to general outlets (green dots in Fig. 2). The main drains were installed in the first few years during irrigation implementation and since then field drains are installed progressively on the initiative of each farmer according to their needs. We have no precise information on the surface that has been equipped with field drains or on their spacing. From field observations, we know that the surface drains were dug at a depth of approximately 2 m, as were the main drains.

Two of these main drains have been equipped with CTD-10 sensors (Meter Group Inc., Pullman, WA, USA) that continuously measure the water level. They collect drainage water from areas of 116 and 2050 hectares each, forming two sub-basins AB1 and AB2 (see Fig. 2). These areas correspond to the topographic basins formed by the main drains at the CTD-10 sensors locations and were computed using the QGIS software with a 2 m resolution DEM provided by the Cartographic and Geological Institute of Catalonia. Table 1 shows for AB, AB1, and AB2,

the area and percentages of the main crop types. Figure 2.b shows the land cover of AB1 and AB2 for 2021.



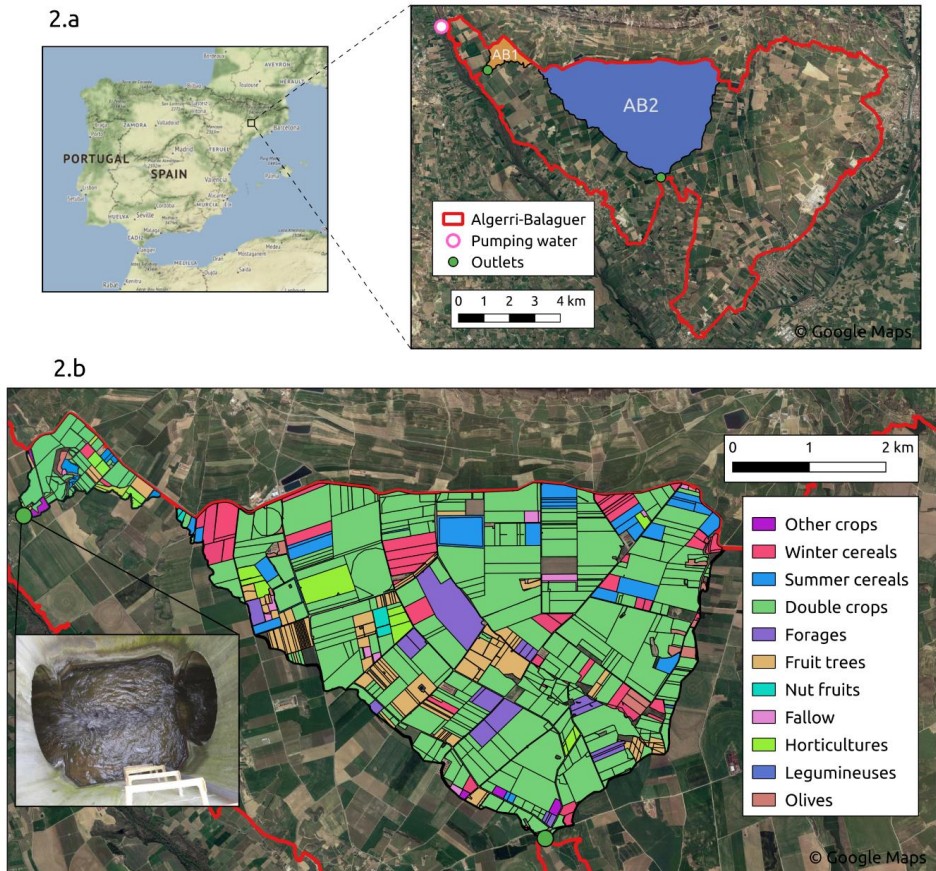

**Figure 2.** The AB irrigation district with the two monitored sub-basins AB1 and AB2 and their outlets, as well as the location of the pumping station for irrigation (coordinates: 797157/4636983.4 ETRS89/UTM zone 31N) (a); Zoom on the two sub-basins with their land use for the year 2021 and a picture of inside the AB1 outlet where water level is measured before being converted into drainage discharge (b).





**Table 1.** Irrigated surfaces of AB, AB1, and AB2, and percentage of surface area occupied by their main crops in 2021 and 2022.

| | Irrigated surface (hectares) | Year | Percentage of surface occupied by... | | |
| --- | --- | --- | --- | --- | --- |
| | | | Double crop (mostly wheat or barley in winter and maize in summer) | Summer cereal (mostly maize) | Others (alfalfa, winter cereals, olives, fruit trees, etc.) |
| AB | 6800 | 2021 | 58 % | 8 % | 34 % |
| | | 2022 | 69 % | 7 % | 24 % |
| AB1 | 116 | 2021 | 59 % | 6 % | 35 % |
| | | 2022 | 69 % | 6 % | 25 % |
| AB2 | 2050 | 2021 | 72 % | 6 % | 22 % |
| | | 2022 | 78 % | 4 % | 18 % |

### 2.1.2 Data

The water levels measured in the AB1 and AB2 outlets are obtained at hourly intervals and converted into daily discharge using the Manning-Strickler equation and the knowledge of the main drain hydraulic characteristics (Altés et al., 2022). The available drainage data used herein cover the period from February 2021 to October 2022 (21 months) for AB1 and from May 2021 to October 2022 (18 months) for AB2. The drainage data of both AB1 and AB2 for the period from May 2021 to October 2022 are presented in Fig. 3, where differences in terms of drainage quantity and dynamics can be observed.

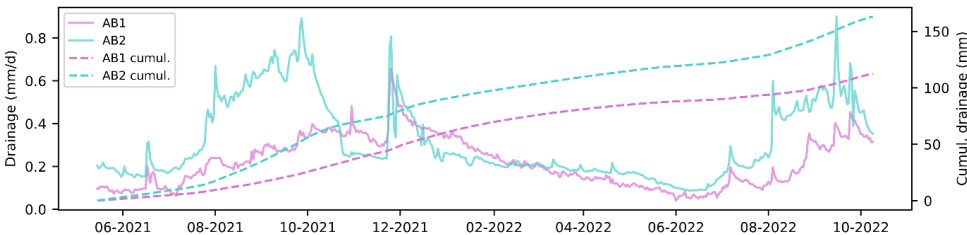

**Figure 3**. Daily drainage data of AB1 and AB2 for the period from May 2021 to October 2022.

The irrigation data consist of the daily flow of water pumped from a river located next to the AB district (see Fig. 2), which is the only supply for the irrigation network. They are provided by the Automatic Hydrological Information System of the Ebro Basin (SAIH). Pumping flow data are aggregated to the weekly scale to consider the potential delay of several days between pumping and application in the field. 5.8% of the volume is removed to account for evaporation loss and leakage, based on a comparison between the water pumped from the river and irrigation data from water meters (Olivera-Guerra et al., 2023).

Soil texture is obtained from the 250-meter resolution SoilGrids product (Hengl et al. 2017; Poggio et al. 2021). It is relatively uniform over the AB area and corresponds to a silty clay loam soil (Jahn et al., 2006).



Meteorological data are obtained from five stations belonging to the Catalan Meteorological Station Network. Two of them are located within the AB district, and the three others are located around the area at a maximum distance of 5 km. The mean standard deviation of the instantaneous measurements of precipitation and $ET_0$ made by the five stations is very low. Therefore, the spatial average of precipitation and $ET_0$ measurements is used as forcing at the scale of AB sub-basins.

**2.2 Description of the four models**

The four models evaluated herein result from the combination of two water balance models (RU and SAMIR) and two drainage discharge models (Reservoir and SIDRA). Their main characteristics are listed in Table 2.

**Table 2.** Description of the four models used.

| | RU | SAMIR | Reservoir | SIDRA |
|---|---|---|---|---|
| Description | Simple water balance model | FAO-2Kc-based crop water balance model | Exponential emptying of a reservoir | Semi-analytical formula derived from the Boussinesq equation |
| Inputs | $ET_0$, rainfall, irrigation, soil texture type | $ET_0$, rainfall, soil texture, land cover, $NDVI$ | Recharge | Recharge, drainage network characteristics, soil texture type |
| Parameters | 2 crop parameters<br>3 soil parameters | 7 crop parameters<br>2 soil parameters<br>2 irrigation parameters | 1 depletion coefficient parameter | 2 drainage network geometry parameters<br>2 soil parameters |
| Sensitivity | 1 param. most sensitive ($S_{inter}$) (Henine et al., 2022; Chelil et al., 2022) | 2 param. most sensitive ($Zr_{max}$ and $a_{Kcb}$) (Laluet et al., 2023) | 1 param. sensitive ($k$) | 2 param. most sensitive ($K$ and $\mu$) (Henine et al., 2022; Chelil et al., 2022) |

**2.2.1 SAMIR**

The SAMIR model (Simonneaux et al., 2009) is a FAO56 double crop coefficient-based model (FAO-2Kc) (Allen et al., 1998) designed to simulate the crop water balance components for daily $ET$ estimation and crop water requirements by considering the plant and soil water status. It uses i) meteorological forcing variables to calculate $ET_0$ (calculated using the Penman-Monteith equation), ii) precipitation, iii) crop and soil parameters to calculate soil reservoir properties, as well as plant and soil resistance to water stress, and iv) Normalized Difference Vegetation Index ($NDVI$) to drive plant development, obtained from the Sentinel-2 satellites with a resolution of 10 m and a revisit of five days.

The daily water balance equation simulated with SAMIR is:

$$Dr_t = Dr_{t-1} + ET_t - P_t - I_t + R_t \qquad (1)$$



where $Dr$ is the root zone depletion, $ET$ the actual evapotranspiration, $P$ the precipitation, $I$ the irrigation, and $R$ the underground recharge. Every term is expressed in mm for the day $t$ (and $t$-$1$ for $Dr$). $ET$ is estimated by multiplying two crop coefficients to $ET_0$ as follows:

$$ET_t = (Kcb_t \cdot Ks_t + Ke_t \cdot Kr_t) \cdot ET_{0_t} \tag{2}$$

where $ET_0 \cdot Kcb \cdot Ks$ is the water transpired by plants ($T$, mm), and $ET_0 \cdot Ke \cdot Kr$ the soil evaporation ($E$, mm). $Kcb$ (-) is the basal crop coefficient governing the potential crop transpiration. It is estimated from a linear relationship with $NDVI$. $Ks$ (-) is the water stress coefficient reducing the potential transpiration, $Ke$ (-) the potential soil evaporation coefficient, and $Kr$ (-) the evaporation reduction coefficient.

$Kr$ is calculated with a pedotransfer function using clay and sand fractions that was derived and evaluated over a variety of sites (Lehmann et al., 2018; Merlin et al., 2016) and recently implemented into SAMIR by Amazirh et al. (2021).

$Ks$ is calculated based on the daily computation of the water balance in the root-zone layer, as follows:

$$Ks_t = \frac{TAW_t - Dr_t}{TAW_t(1-p)} \tag{3}$$

where $Dr$ is calculated from the daily water balance according to Eq. (1), $TAW$ (mm) is the maximum available water in the root zone and $p$ (-) is the fraction of $TAW$ that a crop can extract without facing water stress. Allen et al. (1998) suggests that $p$ controls the water depth threshold below which irrigation should be triggered to avoid crop water stress, by keeping $Dr$ smaller than $TAW \cdot p$ (and thus $Ks$ equal to 1). $TAW$ is estimated as follows:

$$TAW_t = (SM_{FC} - SM_{WP}) \cdot Zr_t \tag{4}$$

where $SM_{FC}$ (m$^3$ m$^{-3}$) is the soil moisture at field capacity and $SM_{WP}$ (m$^3$ m$^{-3}$) the soil moisture at the wilting point, both derived from the soil texture by applying the pedotransfer function proposed by Román-Dobarco et al. (2019). $Zr$ (mm) is the rooting depth that varies between a minimum value (set to 100 mm for annual crops) and a crop-dependent maximum value (reached at the maximum $NDVI$ of the simulated field).

A spatialized version of SAMIR at the plot-scale was recently developed and is used in this study to simulate recharge at AB1 and AB2. More details on the methodology behind this spatialization can be found in Olivera-Guerra et al. (2023).

To simulate plot-scale irrigation using SAMIR, we used the method proposed by Olivera-Guerra et al. (2023), which consists of inverting two time-varying SAMIR irrigation parameters from the irrigation data measured at the pumping station. By applying the inverted parameters to each irrigated field, and by aggregating the resulting simulations over the entire AB district, the simulated irrigation volumes and timing were found quite close to those measured at the pumping station (RMSD < 0.70 mm d$^{-1}$ on average for six irrigation seasons). The values of the SAMIR irrigation parameters found by Olivera-Guerra et al. (2023) for the AB district were used herein for 2021 and 2022. The plot-scale irrigations simulated by SAMIR are then averaged for AB1 and AB2 to be used as forcing in the RU model (as RU is not spatialized).



**2.2.2 RU**

RU is a water balance model designed to simulate the recharge of the water table. It is one of the components of the RU-SIDRA model introduced by Henine et al. (2022). In contrast to SAMIR, RU has not been designed to precisely reproduce $ET$ by simulating plant phenology or processes related to evaporation. Its purpose is rather to reproduce the correct amount of recharge to be converted into drainage with the SIDRA model. While a detailed

description of the model can be found in Henine et al. (2022) and its evaluation in Jeantet et al. (2021), only a general overview is provided here.

RU uses as input precipitation, irrigation, $ET_0$, and soil texture type (for default parameter values). It is composed of a module simulating the net infiltration ($P_{net}$, mm) and a soil reservoir module transforming $P_{net}$ into recharge $R$.

$P_{net}$ is calculated as follows:

$$P_{net_t} = P_t + I_t - CET_t \qquad (5)$$

With $CET$ (mm) being the corrected $ET$, which is computed as follows:

$$CET_t = \begin{cases} ET_{0_t} \cdot e^{\frac{-S_{RFU}-S_t}{S_t}} & if\ S_t < S_{RFU} \\ ET_{0_t} & if\ S_t \geq S_{RFU} \end{cases} \qquad (6)$$

where $S_t$ (mm) is the current water level in the soil reservoir on day $t$ and $S_{RFU}$ (mm) a water level threshold

triggering the plant water stress limiting $ET_0$. $S_{RFU}$ is $0.4 \cdot S_{inter}$ with $S_{inter}$ (mm) being a threshold in the soil reservoir that triggers recharge, analogous to $SM_{FC}$ in SAMIR.

The soil reservoir module is designed to simulate recharge ($R$) depending on three stages related to the amount of water in the soil reservoir ($S$):

$$Stage\ 1: S_t < S_{inter}\ ;\ R_t = 0 \qquad (7)$$

$$Stage\ 2: S_t \in [S_{inter}; S_{max}]\ ;\ R_t = \beta \cdot P_{net_t} \qquad (8)$$

With $S_{max}$ (mm) being the water level below which recharge occurs with a reduction coefficient $\beta$ (−) and calculated as follows:

$$S_{max} = S_{inter} + S_{IDS} \qquad (9)$$

Where $S_{IDS}$ (mm) is the intense drainage season reservoir level (Jeantet et al., 2021). Henine et al. (2022) and Chelil

et al. (2022) found that both $S_{IDS}$ and $\beta$ are not significantly sensitive. Based on the values used in Jeantet et al. (2021), $S_{IDS}$ was set to 20 mm and $\beta$ to 0.33.

$$Stage\ 3: S_t > S_{max}\ ;\ R_t = P_{net_t} \qquad (10)$$



Note that RU is not spatialized, implying that a single simulation is performed for AB1 and AB2 separately with average forcings and parameters.

### 2.2.3 SIDRA

SIDRA is a physically-based model designed to calculate the drainage flow of a drained plot or a drained sub-basin. It is based on the resolution of a semi-analytical formula derived from the Boussinesq equation, which leads to Eq. (11) and Eq. (12). For a complete description of SIDRA, readers are referred to Tournebize et al. (2004), Henine et al. (2022), and Zimmer et al. (2023).

First, the water table level variation under the influence of recharge ($R$) and drainage is computed as follows:

$$\frac{dh_t}{d_t} = \frac{R_t - K\frac{h_t^2}{L^2}}{C\mu} \ ; \ h_{t+1} = h_t + \frac{dh_t}{d_t} \tag{11}$$

where $h$ is the water table (m), $K$ the horizontal hydraulic conductivity (m d$^{-1}$), $\mu$ the drainable porosity (m$^3$ m$^{-3}$), and $C$ a water table shape factor (-) equal to 0.904. $h$ is bounded between 0 and 1.5 (the average drain depth assumed at AB). The drainage of the water table into buried pipes or open drains, leading to drainage flow $Q$, is calculated as follows:

$$Q_t = AK\frac{h^2}{L^2} + (1-A)R_t \tag{12}$$

where $L$ is the half of the drain spacing (or mid-drain spacing, m) and $A$ a water table shape factor (-) equal to 0.896. As we do not have precise information on the field drains location and the proportion of the surface actually equipped with them at AB, $L$ can be either calibrated with drainage data, or set to a value frequently found in the literature (generally comprise between 3 and 12 m).

Boussinesq's equation assumes that buried pipes and open drains rest on an impermeable layer, meaning that the entire water table could be drained after a given period without rain or irrigation. The hydrogeological configuration of the AB district, with the presence of a shallow impervious layer, allows us to assume that this condition is respected.

SIDRA is not spatialized; therefore, when combined with SAMIR (which is spatialized, unlike RU), it uses as input the average daily recharge from all the simulated plots of AB1 and AB2.

### 2.2.4 Reservoir

Reservoir (Cenobio-Cruz et al., 2023) is a conceptual model designed to reproduce the delay between the recharge and the water table draining into a river, buried pipes, or open drains. The idea of this model is that a reservoir filled by recharge is drained according to a linear relationship between the water level $Z$ (mm) and a depletion coefficient $k$ (-). It can be expressed as follows:

$$Q_t = Z_t \cdot k \tag{13}$$



Cenobio-Cruz et al. (2023) incorporated a reservoir size parameter to simulate overflow and generate quick flows. We decided not to include this parameter to make the Reservoir model as simple as possible.

### 2.2.5 Initialization of the state variables

The four models RU-Reservoir, RU-SIDRA, SAMIR-Reservoir, and SAMIR-SIDRA require initialization of their state variables. These variables were initialized with a 12-month spin-up simulation using data from 2020. SAMIR initializes the depletion parameters of the soil and surface reservoirs, RU the water level in the soil reservoir, SIDRA the water table level, and Reservoir the water level in the conceptual reservoir.

### 2.2.6 Sensitivity of model parameters

Laluet et al. (2023) conducted an extensive sensitivity analysis of *ET* and recharge simulated by SAMIR over a wide range of agro-pedoclimatic conditions. Two of the nine parameters were found to dominate the model sensitivity: $a_{Kcb}$ governing the relationship between *NDVI* and *Kcb* (related to *T* demand), and $Zr_{max}$ being the maximum rooting depth governing the size of the root zone reservoir.

Henine et al. (2022) and Chelil et al. (2022) analyzed the sensitivity of the RU-SIDRA parameters for drainage simulation on two different plots. They both showed that for the RU model, the $S_{inter}$ parameter controls most of the model sensitivity, and that for SIDRA it is $K$ and $\mu$.

### 2.3 Strategy for evaluating the models' precision

It is reminded that by precision it is meant the ability of a model to approximate the observed drainage data as closely as possible by means of site-specific calibration (using drainage measurements). The most sensitive parameters of the four models were calibrated using an automatic calibration algorithm. Since we do not have information on the mid-drain spacing on AB, we calibrated the *L* parameter, bringing the number of parameters to be calibrated for RU-SIDRA to four, and for SAMIR-Reservoir to five. Indeed, although Chelil et al. (2022) showed that *L* is not very sensitive when it varies between 3.5 m and 6 m, its uncertainty within the AB district is large enough for it to be significantly sensitive. The other less sensitive parameters are fixed at the default values given in the literature. In an effort to analyze the variability of the parameters from one hydrological year to the next, we split the data into a 12-month period named "2021 period", and a period with the remaining months (9 months for AB1, 6 months for AB2) named "2022 period".

The calibration method used is the multi-objective Non-dominated Sorting Genetic Algorithm (NSGA-II) (Deb et al., 2002). For the case of SAMIR, which simulates both irrigation and recharge, we use a multi-objective method to ensure that the calibration of $a_{Kcb}$ and $Zr_{max}$ parameters does not significantly modify the simulated irrigation. Therefore, for both SAMIR-Reservoir and SAMIR-SIDRA, drainage and irrigation are optimized together, whereas for both RU-Reservoir and RU-SIDRA, only drainage is optimized (an averaged irrigation is given as forcing in this case as RU cannot simulate irrigation). NSGA-II is one of the most widely used multi-objective algorithms. It implements a fast non-dominant sorting approach to discriminate solutions based on the concept of dominance and Pareto optimality. It provides a set of optimal non-dominated solutions (set of parameters), allowing the user to choose the best solution according to his priorities. In the SAMIR case, the best solution would



be the one that simulates the most precise drainage, provided that it simulates irrigation consistent with the observed data at the pumping station. Readers are referred to Deb et al. (2002), Bekele and Nicklow (2007), and

Shafii and De Smedt (2008) for a detailed description of the algorithm.

NSGA-II requires a distribution provided by the user for each calibrated parameter. The distribution and references used for each model are provided in Table 3. For the *K* and *μ* parameters of SIDRA, the same distribution as in Jeantet et al. (2021) is used because they are derived from field measurements performed on 15 silty soils, being similar to the soil texture of AB1 and AB2. For the *k* parameter of Reservoir, we use a normal distribution with a

mean of 0.02, as this is the average value obtained by Cenobio-Cruz et al. (2023) on approximately 25 watersheds in northern Spain.

For drainage calibration, the objective function Kling-Gupta Efficiency (KGE) (Gupta et al., 2009) is used:

$$KGE = 1 - \sqrt{(r-1)^2 + (\alpha - 1)^2 + (\delta - 1)^2}$$    (14)

where *r* is the Pearson's correlation coefficient, *α* the bias component, and *δ* the ratio of the discharge variance:

$$\alpha = \frac{m_s}{m_o} \ \ and \ \ \delta = \frac{\sigma_s}{\sigma_o}$$    (15)

where *m* and *σ* are the mean and standard deviation, respectively. Subscripts *s* and *o* represent the simulated and observed flow, respectively.

Since KGE tends to place more weight on high flows (Santos et al., 2018), we chose to use KGE ($Q^{0.5}$). KGE ($Q^{0.5}$) is the KGE calculated from the squared roots of simulated and observed drainage, allowing the weights between high and low flows to be more balanced. Following Jeantet et al. (2021), we consider simulations as "excellent"

when KGE ($Q^{0.5}$) is larger than 0.8, "very good" when it is larger than 0.7, "good" when it is larger than 0.6, "acceptable" when it is between 0.5 and 0.6, and "unsatisfactory" when it is below 0.5. Furthermore, to get a reference in mind, a KGE of -0.41 is equivalent to having a simulation performance equal to the average of the observed data (Knoben et al., 2019).

The RMSD objective function is used for irrigation calibration:

$$RMSD = \sqrt{\frac{\sum_{i=1}^{j}(\hat{y}_i - y_i)^2}{j}}$$    (16)

where *j* is the number of days in the simulated time series, *i* is one day of the time series, $\hat{y}_i$ is the simulated time series, and $y_i$ is the reference time series. We consider irrigation to be well simulated when the RMSD calculated between the irrigation measured at the pumping station and the one simulated by SAMIR for all the plots in AB is

below 0.70 mm d$^{-1}$. This value corresponds to the RMSD found on average from 2017 to 2021 by Olivera-Guerra et al. (2023).



**Table 3.** Distributions of the main parameters calibrated with the NSGA-II algorithm to evaluate the models' precision, and associated references.

| | Parameter | Distribution | Reference |
|---|---|---|---|
| SAMIR | $a_{Kcb}$ (-) | Normal (mean: 1.45; std. dev.: 0.12) | Laluet et al. (2023) |
| | $Zr_{max}$ (mm) | Normal (mean: 1000; std. dev.: 170) | Allen et al. (1998) Pereira et al. (2021) |
| RU | $S_{inter}$ (mm) | Normal (mean: 138; std. dev.: 53) | Jeantet et al. (2021) |
| SIDRA | $K$ (m d$^{-1}$) | Lognormal (mean: 0.99; std. dev.: 2.53) | |
| | $u$ (-) | Lognormal (mean: 0.018; std. dev.: 2.19) | |
| | $L$ (m) | Uniform (low: 4; high: 60) | |
| Reservoir | $k$ (-) | Normal (mean: 0.02; std. dev.: 0.05) | Cenobio-Cruz et al. (2023) |


**2.4 Strategy for evaluating the models' accuracy**

Complementarily to the evaluation of the models' precision, this study also aims to assess the accuracy of the four models. It is reminded that by accuracy we mean the ability of a model to approximate the observed data as closely as possible, by relying only on default values given by the literature for its main parameters, i.e., without any site-
specific calibration step.

To this end, for each of the four models, 2000 sets of their most sensitive parameters are generated randomly by means of a Monte Carlo sampling, using the distributions presented in Table 3. Unlike the previous step, here the value of the mid-drain spacing $L$ is set to 6 m since it is a value frequently found in the literature. In addition, in order to focus only on the drainage accuracy evaluation, the irrigation obtained with SAMIR during the precision
evaluation step is injected into SAMIR as a forcing. The KGE ($Q^{0.5}$) obtained with a simulation performed with average default parameters is calculated, and the ensemble generated by the 2000 Monte-Carlo simulations is analyzed.

**3 Results and discussion**

**3.1 Precision evaluation**

In this section we will first give a quick overview of the irrigation simulated by SAMIR and used as a forcing by RU. The precision of the drainage simulated by each model is then presented. Finally, we will explain why the model precision differs between the four models and provide some recommendations and perspectives.

**3.1.1 Irrigation simulated by SAMIR**

Table 4 shows for SAMIR-Reservoir and SAMIR-SIDRA the RMSD obtained between simulated and observed irrigation at the pumping station over all the irrigated plots of AB, resulting from the NSGA-II multi-objective



calibration performed for AB1 and AB2. The average RMSD obtained is 0.35 mm d$^{-1}$ for 2021 and 0.66 mm d$^{-1}$ for 2022, being in line with the quality criteria defined previously (< 0.70 mm d$^{-1}$). The average amount of irrigation simulated for the AB1 sub-basin is 592 mm on average for the period from May to October 2021, and 693 mm for the period from May to October 2022. For AB2, it is 619 mm in 2021 and 720 mm in 2022. These amounts are fully consistent given the amounts of irrigation measured at the pumping station and the proportions of surface used for double crops and summer cereals in each of the AB1 and AB2 sub-basins compared to those in the entire AB district (see Table 2).

**Table 4.** RMSD between daily simulated and observed irrigation, and seasonal (between May and October for each year) cumulated simulated irrigation obtained with site-calibrated SAMIR-Reservoir and site-calibrated SAMIR-SIDRA separately.

| | Irrigation obtained for AB1 calibration | | | | Irrigation obtained for AB2 calibration | | | |
|---|---|---|---|---|---|---|---|---|
| | 2021 period | | 2022 period | | 2021 period | | 2022 period | |
| | RMSD (mm d$^{-1}$) | Average amount (mm) | RMSD (mm d$^{-1}$) | Average amount (mm) | RMSD (mm d$^{-1}$) | Average amount (mm) | RMSD (mm d$^{-1}$) | Average amount (mm) |
| SAMIR-Reservoir | 0.37 | 595 | 0.69 | 693 | 0.34 | 621 | 0.62 | 719 |
| SAMIR-SIDRA | 0.32 | 590 | 0.65 | 694 | 0.38 | 617 | 0.66 | 720 |

### 3.1.2 Which model is more precise?

Figures 4 and 5 compare the simulated and observed drainage for the AB1 and AB2 sub-basins, respectively, and the 2021 and 2022 periods. Table 5 shows that among the 16 combinations of model/sub-basin/period, nine have KGE (Q$^{0.5}$) considered as "excellent", four "very good", two "good", and one "acceptable", and there is no "unsatisfactory" simulation. RU-Reservoir stands out as the most precise model (mean KGE (Q$^{0.5}$) = 0.87), followed by SAMIR-Reservoir (0.79), RU-SIDRA (0.76), and SAMIR-SIDRA (0.68). From these results, two highlights stand out:

i) The models based on Reservoir are more precise in terms of drainage simulations than those based on SIDRA. This is particularly true for the AB1 sub-basin.

ii) The models based on RU are more precise in terms of drainage simulations than those based on SAMIR.



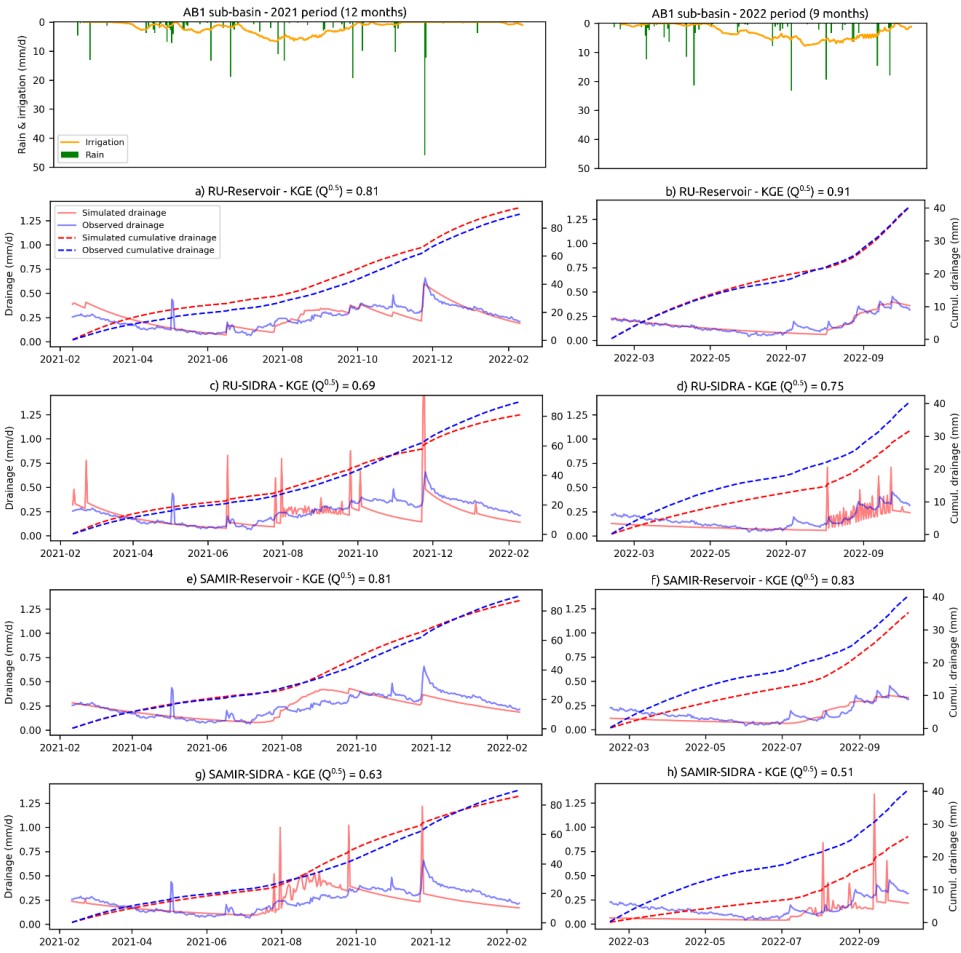

**Figure 4.** Daily and cumulated drainage of the AB1 sub-basin simulated by the four site-calibrated models for 2021 (left) and 2022 (right) periods. Plots at the top show the observed precipitation and the simulated irrigation.





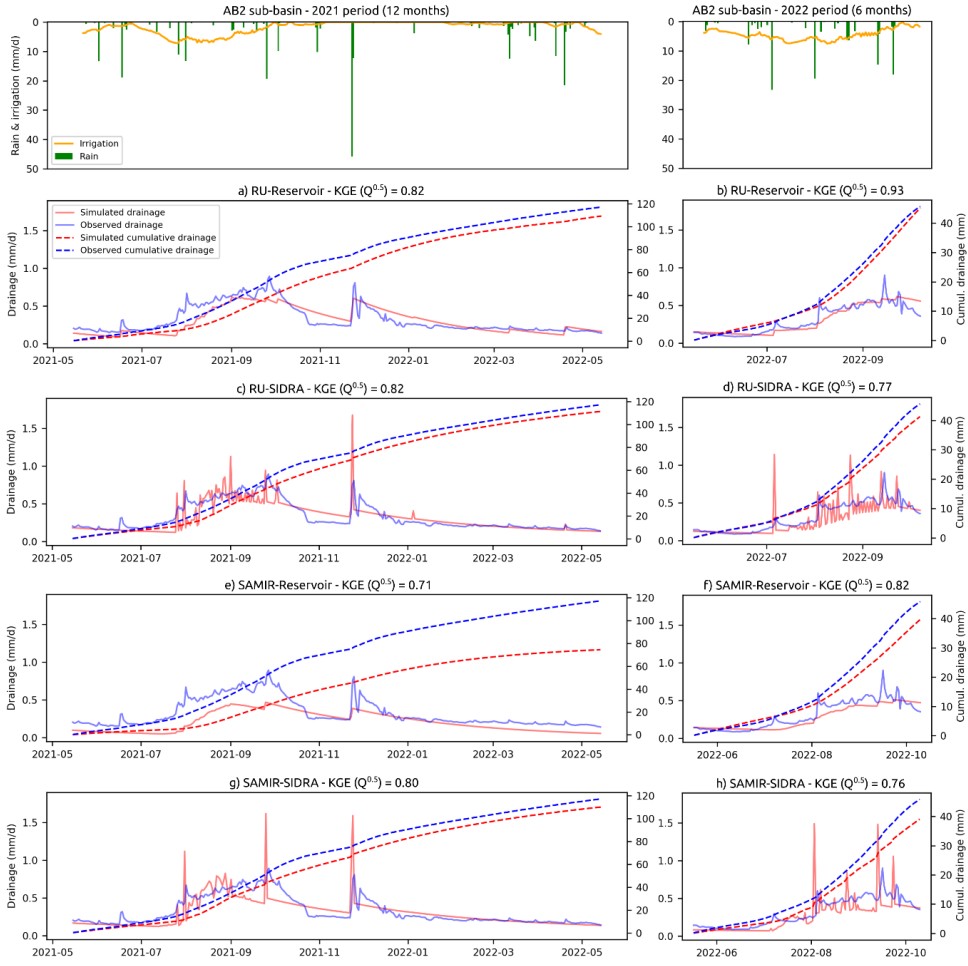

**Figure 5.** Same as Fig. 4 but for AB2 sub-basin.

410




**Table 5.** KGE ($Q^{0.5}$) obtained with NSGA-II calibration for the AB1 and AB2 sub-basins, the four models, and both study periods. Values above 0.8 are considered as "excellent", between 0.7 and 0.8 as "very good", between 0.6 and 0.7 as "good", and between 0.5 and 0.6 as "acceptable".

|  | KGE ($Q^{0.5}$) for AB1 | | KGE ($Q^{0.5}$) for AB2 | |
|---|---|---|---|---|
|  | 2021 period | 2022 period | 2021 period | 2022 period |
| RU-Reservoir | 0.81 | 0.91 | 0.82 | 0.93 |
| RU-SIDRA | 0.67 | 0.75 | 0.82 | 0.77 |
| SAMIR-Reservoir | 0.81 | 0.83 | 0.71 | 0.82 |
| SAMIR-SIDRA | 0.63 | 0.51 | 0.80 | 0.76 |

415

**Table 6.** Parameter values obtained with NSGA-II calibration for the AB1 and AB2 sub-basins, the four models, and both study periods.

|  | Parameters | AB1 / 2021 period | AB1 / 2022 period | AB2 / 2021 period | AB2 / 2022 period |
|---|---|---|---|---|---|
| RU-Reservoir | $S_{inter}$ (mm) | 13 | 76 | 16 | 27 |
|  | $k$ (-) | 0.015 | 0.008 | 0.014 | 0.007 |
| RU-SIDRA | $S_{inter}$ (mm) | 10.5 | 94 | 28 | 51 |
|  | $K$ (m d$^{-1}$) | 0.58 | 1.30 | 1.08 | 0.75 |
|  | $\mu$ (-) | 0.07 | 0.13 | 0.14 | 0.11 |
|  | $L$ (m) | 29 | 42 | 38 | 39 |
| SAMIR-Reservoir | $a_{Kcb}$ (-) | 1.34 | 1.23 | 1.29 | 1.23 |
|  | $Zr_{max}$ (mm) | 1087 | 1130 | 1087 | 1004 |
|  | $k$ (-) | 0.009 | 0.004 | 0.012 | 0.005 |
| SAMIR-SIDRA | $a_{Kcb}$ (-) | 1.36 | 1.19 | 1.23 | 1.24 |
|  | $Zr_{max}$ (mm) | 1072 | 1104 | 943 | 1079 |
|  | $K$ (m d$^{-1}$) | 0.55 | 0.58 | 0.91 | 0.70 |
|  | $\mu$ (-) | 0.10 | 0.10 | 0.17 | 0.09 |
|  | $L$ (m) | 36 | 38 | 32 | 43 |





### 3.1.3 Why are Reservoir-based models more precise?

The better performance of the site-calibrated RU-Reservoir and SAMIR-Reservoir models is explained by the differences in the formalism of Reservoir and SIDRA and by the low responsiveness of the AB1 and AB2 hydrosystems. Indeed, SIDRA was designed to simulate flow peaks followed by relatively steep recession curves. In contrast, Reservoir does not simulate peaks, and generates flow according to a depletion coefficient $k$ that can be very low. However, the measured drainage dynamics for AB1 and AB2 are representative of low-responsive hydrosystems. This can be seen in Fig. 4 and 5 where the 40 mm rainfall in December 2021 generates a peak of less than 1 mm d$^{-1}$ for both sub-basins, followed by a recession curve being quite smooth with a discharge that never reaches 0 during the hydrological year. In comparison, the data used by Jeantet et al. (2021) from 22 French experimental sites, accounting for nearly 200 hydrological years, show winter peaks exceeding 20 mm d$^{-1}$ in most of the studied years, followed by steep recession curves where 0 flow is reached in a few weeks. Therefore, the Reservoir model is favored by the low responsiveness of the AB1 and AB2 hydrosystems, and this is especially true for AB1. The SIDRA model shows a better precision for AB2 than for AB1 because AB2 is more responsive with larger amounts of discharge. In addition, the values of the Reservoir depletion coefficient $k$ are low at our sites (0.009 on average) compared to those obtained by Cenobio-Cruz et al. (2022) (0.02 on average), which again reflects the relatively low responsiveness of the studied area.

This low responsiveness is not explained by the soil, since the mean calibrated values of $K$ and $\mu$ (0.81 m d$^{-1}$ and 0.11 m$^3$ m$^{-3}$, respectively; see Table 6) are consistent with the order of magnitude found in Jeantet et al. (2021) for a similar soil type ($K$ from 0.1 to 1.8 m d$^{-1}$, and $\mu$ from 0.05 to 0.08 m$^3$ m$^{-3}$). The explanation seems to lie in the fact that the surfaces of AB1 and AB2 are not fully equipped with field drains. On non-equipped plots, the transfer time would be longer from the recharge location to the main drain. The values of the calibrated mid-drain spacing $L$ support this assumption, with an average of 37 m (see Table 6), being a value that could represent an average between low $L$ values for plots equipped with field drains, and high $L$ values for plots that are not.

### 3.1.4 Why are RU-based models more precise?

To understand the better performance of the site-calibrated RU-Reservoir and RU-SIDRA models, in comparison with SAMIR-Reservoir and SAMIR-SIDRA, respectively, it is again necessary to look at the differences in formalism between RU and SAMIR. Indeed, RU, unlike SAMIR, has a stage during which a reduction factor $\beta$ of 0.33 is applied to the recharge (Eq. (8), this stage is triggered with the water level in the soil reservoir). In AB1 and AB2, this stage is triggered in particular during the irrigation season and results in the spread of recharge amounts. This benefits RU-SIDRA, which, owing to the lower recharge events simulated by RU, generates lower peaks consistent with the drainage observations. This process is well illustrated in Fig. 4.d, where RU-SIDRA simulates numerous small peaks during the irrigation period, allowing better matching of the observations, whereas in Fig. 4.h, SAMIR-SIDRA simulates larger and fewer peaks.

Furthermore, the $S_{inter}$ values of RU obtained through calibration are very low (17 mm on average for the 2021 period and 61 mm for the 2022 period; see Table 5) compared with those obtained by Jeantet et al. (2021) (138 mm on average). This is due to the fact that the RU-based models are not spatialized and use an average irrigation



derived from the irrigation simulated by SAMIR whether for intensely irrigated corn plots generating a lot of recharge, or for non-irrigated plots generating no recharge. Because they simulate an average plot using average irrigation, RU-based models simulate less recharge. To compensate for this, the NSGA-II optimization algorithm

finds low $S_{inter}$ values in order to reduce the reservoir size, and therefore the $ET$, resulting in increasing the recharge. This explains the low $S_{inter}$ values retrieved over AB1 and AB2.

### 3.1.5 Recommendations and perspectives

Based on the results obtained over AB by the four models with varying complexity levels, we recommend the use of RU-Reservoir when drainage data are available for calibration. The simplest model can reproduce the drainage

observed at both the AB1 and AB2 outlets fairly well. In fact, the RU-Reservoir model efficiently combines the performance of RU (allowing a better temporal distribution of recharge than SAMIR) and Reservoir (offering drainage simulations with relatively low responsive dynamics), which fits perfectly with the present study. However, if a study is concerned with a more responsive hydrosystem with larger peaks and steeper recession curves, we would recommend the use of RU-SIDRA. Indeed, for the AB2 sub-basin, which is slightly more

responsive than AB1, RU-SIDRA shows during the 2021 period (Fig. 5.c) a precision comparable to that of RU-Reservoir (Fig. 5.a). This leads us to assume that RU-SIDRA could be more appropriate than RU-Reservoir for even more responsive hydrosystems.

Since RU is not spatialized and has a very simple formalism, the models based on it are simpler to run and require fewer resources than those based on SAMIR. However, they do not take into account the spatial heterogeneity in

terms of irrigation generally encountered in irrigated sub-basins, resulting in very low $S_{inter}$ calibrated values in order to simulate enough recharge. A spatialization of the RU-based models with irrigation data for each plot would lead to higher calibrated $S_{inter}$ values more consistent with those proposed in Jeantet et al. (2021). However, RU, unlike SAMIR, cannot simulate irrigation and its spatialization would then require plot-scale irrigation data that are rarely available.

Note that in some cases, SAMIR shows a relatively low KGE ($Q^{0.5}$) and difficulties in reproducing the right amount of drainage (see Fig. 4.d, 4.h, and 5.e). Modifying the SAMIR formalism by taking inspiration from RU and adding a stage related to the soil water availability in which recharge is limited by a factor $\beta$ could help improve the precision of SAMIR-SIDRA and SAMIR-Reservoir.

### 3.2 Accuracy evaluation

Figures 6 and 7 show for each of the 16 model/sub-basin/period combinations the drainage simulated by using the mean values of default parameters as input to each model (red line), and the 2000 model runs from randomly generated input parameters sets within pre-defined distributions provided by the literature (gray lines). Table 7 summarizes the KGE ($Q^{0.5}$) obtained with the average default parameters. It indicates that all the 16 cases present "unsatisfactory" KGE ($Q^{0.5}$) values (below 0.5). Moreover, nine cases show KGE ($Q^{0.5}$) lower than -0.41, the value

corresponding to the KGE obtained with the temporal average of the observed data.

RU-Reservoir shows relatively satisfactory KGE ($Q^{0.5}$) for the 2022 period (0.22 for AB1 and 0.29 for AB2). However, when looking at the drainage dynamics and amounts illustrated in Fig. 6.b and 7.b it appears that these



performances are due to the nature of the objective function KGE ($Q^{0.5}$) giving a significant importance to low flows. SAMIR-Reservoir shows a relatively good KGE ($Q^{0.5}$) for AB2 for the period 2022 (0.29). Furthermore, the timing and quantities that SAMIR-Reservoir simulated for AB1 for both periods 2021 and 2022, as well as for AB2 for the 2021 period, are more consistent than those simulated by the other three models.

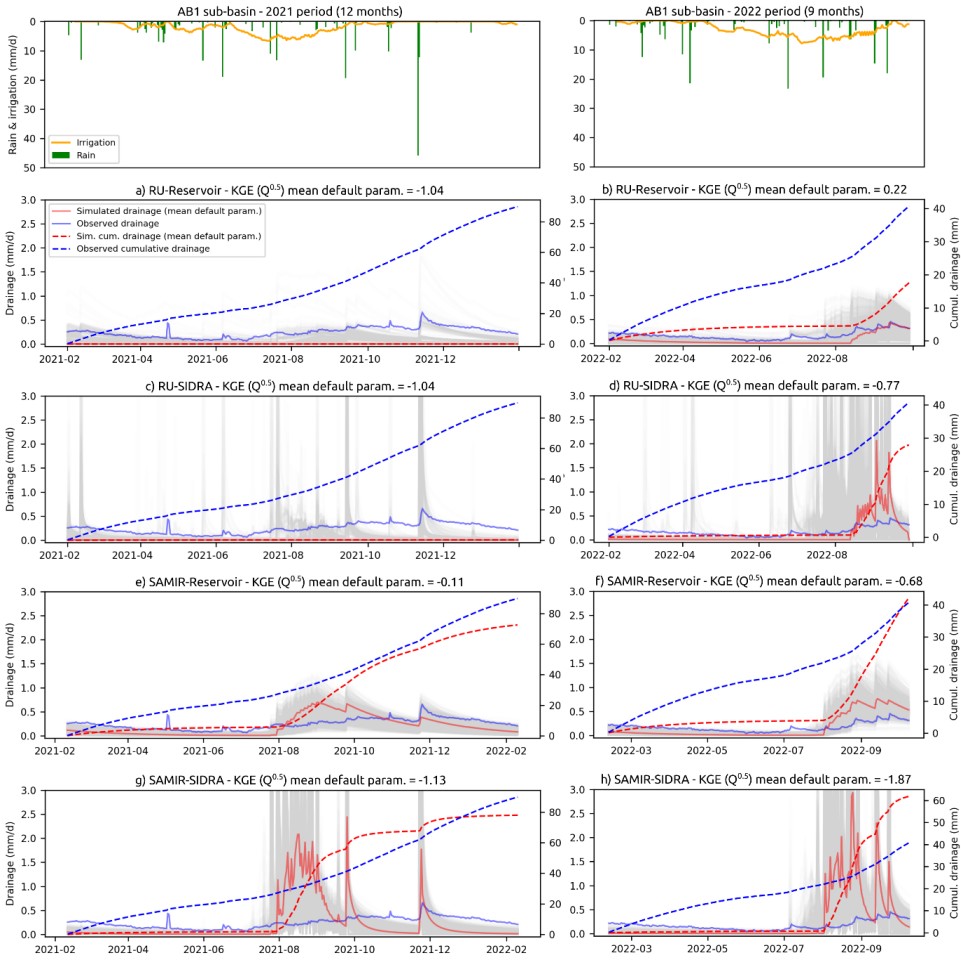

**Figure 6.** Daily drainage of AB1 sub-basin simulated using average default parameters of the four models separately (red line), and the drainage values ensemble obtained by running each model 2000 times using randomly generated input parameters sets within pre-defined distributions (gray lines).



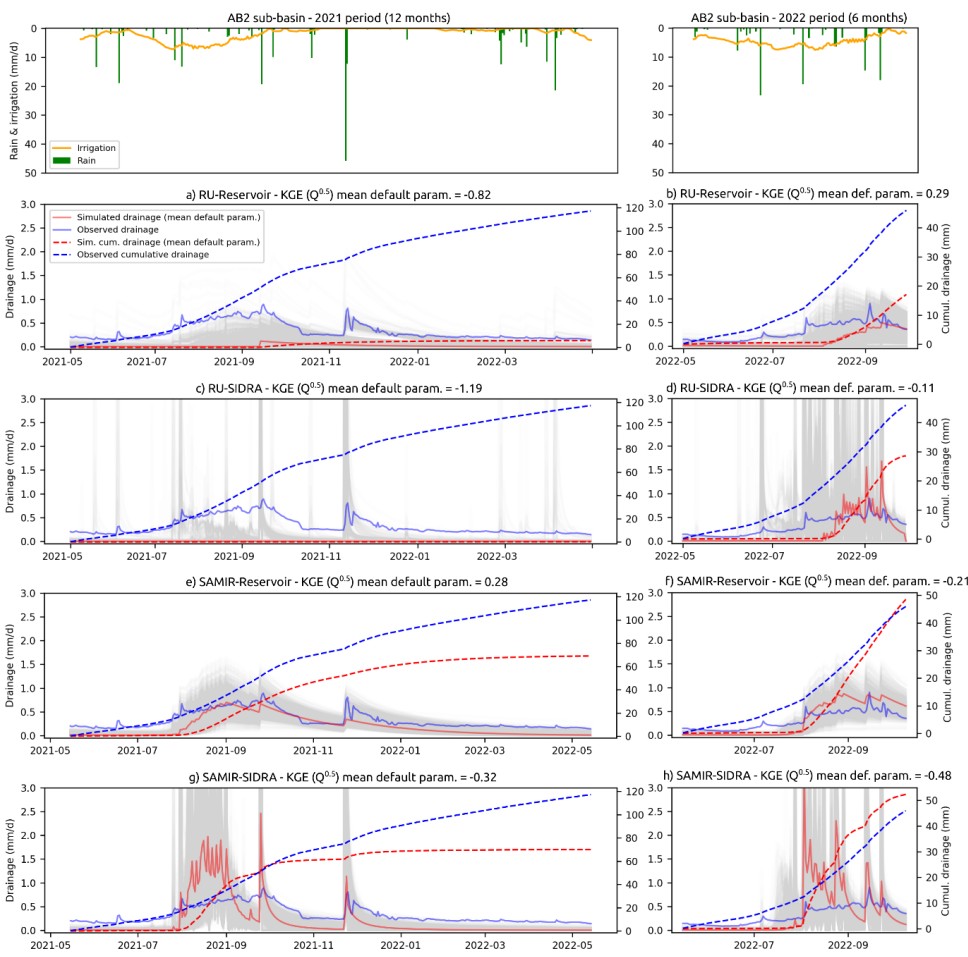

**Figure 7.** Same as Fig. 6 but for the AB2 sub-basin.





**Table 7.** KGE ($Q^{0.5}$) values obtained with average default values of model parameters for the AB1 and AB2 sub-basins, the four models, and the two study periods separately.

| | KGE ($Q^{0.5}$) for AB1 | | KGE ($Q^{0.5}$) for AB2 | |
|---|---|---|---|---|
| | 2021 period | 2022 period | 2021 period | 2022 period |
| RU-Reservoir | -1.04 | 0.22 | -0.82 | 0.29 |
| RU-SIDRA | -1.04 | -0.77 | -1.19 | -0.11 |
| SAMIR-Reservoir | -0.11 | -0.68 | 0.28 | -0.21 |
| SAMIR-SIDRA | -1.13 | -1.87 | -0.32 | -0.48 |

### 3.2.1 Accuracy of RU-Reservoir and RU-SIDRA

Figures 6.a, 6.c, 7.a, and 7.c show that the RU-based models do not simulate any discharge for the 2021 period
with the average default parameters (red lines). This is related to the fact that, in a context where RU is not spatialized while irrigation is spatially heterogeneous, the optimal $S_{inter}$ value to generate sufficient recharge is lower than those given in the literature. The $S_{inter}$ values taken from the literature for the accuracy evaluation are then too high to simulate enough recharge. For the 2022 period, relatively small amounts of drainage discharge are however simulated during the irrigation period because the irrigation amounts applied in 2022 are significantly
larger than in 2021 (about 100 mm more for the period from May to October).

### 3.2.2 Accuracy of SAMIR-SIDRA

Figures 6 and 7 show that with average default parameter values, the SAMIR-based models simulate the drainage dynamics with some consistency with the irrigation season and the rain events, for both 2021 and 2022 periods. However, SAMIR-SIDRA shows a lower performance than SAMIR-Reservoir. Several factors may explain this
result. First, although they are physical parameters, the values of the SIDRA parameters $K$ and $\mu$ found in the literature do not necessarily correspond to their optimal values for a given site. Second, SIDRA is more appropriate for more responsive hydrosystems than AB1 and AB2. Finally, the lack of information on the mid-drain spacing $L$ on the AB district led us to set it at 6 m, whereas when calibrated on AB1 and AB2 this parameter is on average 37 m (considering the plots not equipped with field drains). This lower $L$ value results in a high reactivity of the
simulated drainage.

### 3.2.3 Accuracy of SAMIR-Reservoir

Figures 6.e, 6.f, 7.e, and 7.f. show that the drainage simulated by SAMIR-Reservoir with average default parameter values is more consistent with the observed drainage than the other three models. It also presents less variability within the 2000 simulations ensemble. Fig.7.e shows that SAMIR-Reservoir simulates particularly well the
drainage dynamics and quantities for the AB2 sub-basin during the 2021 period. For AB1 during the 2021 and 2022 periods, and for AB2 during the 2022 period, when the discharge is lower, SAMIR-Reservoir tends to overestimate the drainage.



The SAMIR-Reservoir accuracy varies spatially between AB1 and AB2 and temporally between the 2021 and 2022 periods. These differences reflect the semi-empirical nature of SAMIR and Reservoir models. Indeed, they can be attributed to the impact of unrepresented processes (e.g., lateral flows) or misrepresentation of *ET* and recharge processes in unusual situations (e.g., 2022 drought and heat waves). Especially the lateral flow is fully neglected by SAMIR. Such subsurface flows may come from outside sub-basins boundaries and contribute significantly to the sub-basin discharge measured at the outlet, depending on the hydrometeorological conditions encountered in a given year. Note that representing subsurface lateral flows would require more complex models than those tested in this study and additional data (e.g., piezometric), which are currently unavailable in the study area.

### 3.2.4 Recommendations and perspectives

Due to the semi-empirical nature of the four models investigated in this study, it is difficult to reproduce the drainage discharge with default parameters from the literature. A limitation of RU-based models is their lack of spatialization, leading to the use of an average irrigation in forcing, while the irrigation of AB1 and AB2 sub-basins is spatially heterogeneous. This results in RU not having enough irrigation to generate a correct amount of recharge with the $S_{inter}$ values suggested in the literature (being too high). One way to overcome this would be to spatialize RU with irrigation data at each plot, but these data are rarely available.

SAMIR-Reservoir offers a certain consistency with the observed data and could then provide an approximate idea of the drainage dynamics and amount occurring in an ungauged irrigated sub-catchment. A way to improve the robustness of SAMIR-Reservoir could be to investigate a link between the *k* parameter and the soil texture or the characteristics of the drainage network (drain depth, mid-drain spacing, and drained surface).

### 4 Conclusion

Estimating the drainage in semi-arid irrigation conditions is of great importance to prevent soil salinization issues, and to assess the water productivity and the irrigation impact on downstream ecosystems. A few studies have used physically-based models to simulate drainage in irrigated context. However, these models have many parameters requiring extensive data that are rarely available. In this paper, we thus assessed the capacity of four parsimonious semi-empirical models to simulate drainage at the scale of two sub-basins of the AB district, northeastern Spain. The four models are built from the combination of two surface models (RU and SAMIR) and two subsurface models (Reservoir and SIDRA) with varying complexity levels: RU-Reservoir (two main parameters), RU-SIDRA (three main parameters), SAMIR-Reservoir (three main parameters) and SAMIR-SIDRA (four main parameters). SAMIR is based on the FAO-56 *ET* and crop water balance formulations relying on two main sensitive parameters, while RU is a simplified version of the FAO-56 relying on a single sensitive parameter only. SIDRA solves the Boussinesq equation from two main sensitive parameters, while Reservoir is an empirical drainage model based on a single depletion parameter only.

The precision of the four models, i.e., their ability to reproduce observed drainage data with a site-specific calibration, was first evaluated. An optimal calibration approach was implemented for each model and each sub-



basin using the multi-objective genetic algorithm NSGA-II. The comparison between the drainage simulated by
site-specific calibrated models and observations indicates that RU-Reservoir presents a better precision, followed
closely by SAMIR-Reservoir. This is explained by the fact that Reservoir model is well suited to represent the low
responsiveness of both studied sub-basins, and that RU model manages to artificially better spread out the recharge
events during the irrigation period than SAMIR model.

The point is that the drainage observations required to undertake a site-specific calibration are rarely available.
Therefore, the accuracy of the four models was also evaluated. By accuracy, it is meant the ability of the four
models to reproduce the observed drainage using default parameter values provided by the literature. The
comparison between the drainage simulated by default-calibrated models and observations indicates that SAMIR-
Reservoir is the only model among the four tested capable of providing a rough estimate of the drainage dynamics
and amounts from default parameters.

However, it was found that the accuracy of SAMIR-Reservoir is quite variable from one sub-basin to another and
from one hydrological year to another. Therefore, calibration strategies are still needed to reduce uncertainties in
SAMIR-Reservoir drainage estimates in sub-basins with contrasted conditions. In addition, better constraining the
value of the Reservoir's depletion coefficient $k$, especially by seeking a link with soil texture, or with the
characteristics of the drainage network, should be investigated in future studies to improve the accuracy of SAMIR-
Reservoir.

Furthermore, our study took place in an irrigation district where the water use is known and accurately monitored
through the pumping data. Such irrigation data are rarely available in practice, and no model can predict drainage
accurately based on inaccurate irrigation forcing, regardless of the model calibration issue. It is hence crucial to
develop tools to retrieve the irrigation practices, notably at the integrated spatial scales of sub-basin or irrigation
district. To this end, many recent works seek to assimilate satellite products of soil moisture or $ET$ in crop water
balance models at a range of scales (e.g., Ouaadi et al., 2021; Massari et al., 2021; Dari et al., 2022; Olivera-Guerra
et al., 2020). The coupling of such remote sensing approaches with surface and subsurface models is likely to
improve the predictive capabilities of drainage on irrigated areas.

*Code availability.* All Python codes will be provided by the corresponding author upon request. The Python
package spotpy (Houska et al., 2015) used for the NSGA-II multi-objective optimization is available at
https://github.com/thouska/spotpy.

*Data availability.* Drainage data are not publicly available. Daily precipitation and $ET_0$ data were provided by the
Meteorological Service of Catalonia and are available at https://ruralcat.gencat.cat/agrometeo.estacions (accessed
2 March 2023). The 2 m resolution DEM was provided by the Institute of Cartography of Catalonia and is available
at https://www.icgc.cat/ca/Descarregues/Elevacions/Model-d-elevacions-del-terreny-de-2x2-m (accessed 2
March 2023). Crop type information was provided by the Department of Climate Action, Food, and Rural Agenda
of the Region of Catalonia and is available at https://agricultura.gencat.cat/ca/ambits/desenvolupament-
rural/sigpac/descarregues (accessed 2 March 2023). SoilGrid products are available at https://maps.isric.org
(accessed 2 March 2023).



*Author contributions.* PL and OM conceptualized the work. PL, OM, LOG, and PQS provided the methodological guidelines. VA and JMV collected data. PL, VR, and LOG developed the code. PL drafted the paper. PL, OM, LOG, VR, VA, JMV, AJ, JT, PQS, ABO, and OCC all revised the paper and contributed to its analyses and discussions.

*Competing interests.* The authors declare that they have no conflict of interest.

*Acknowledgment.* The authors would like to thank the Comunitat de Regants Canal Algerri Balaguer and the Automatic Hydrological Information System of the Ebro Basin for providing the irrigation observation data used in this study. We also want to thank Eric Chavanon (CESBIO) who helped to optimize the SAMIR code.

*Financial support.* This study was supported by the IDEWA project (ANR-19-P026-003) of the Partnership for
research and innovation in the Mediterranean area (PRIMA) program and by the Horizon 2020 ACCWA project (grant agreement # 823965) in the context of the Marie Sklodowska-Curie research and innovation staff exchange (RISE) program.

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
