# Peer review of "Drainage assessment of irrigation districts: on the precision and accuracy of four parsimonious models"

_EGUsphere, 2023_

## Author Comment (AC1)

| | Irrigation obtained for AB1 calibration | | | | Irrigation obtained for AB2 calibration | | | | In situ irrigation (entire AB district) | |
| --- | --- | --- | --- | --- | --- | --- | --- | --- | --- | --- |
| | 2021 | | 2022 | | 2021 | | 2022 | | 2021 | 2022 |
| | RMSD (mm d$^{-1}$) | Average amount (mm) | RMSD (mm d$^{-1}$) | Average amount (mm) | RMSD (mm d$^{-1}$) | Average amount (mm) | RMSD (mm d$^{-1}$) | Average amount (mm) | Average amount (mm) | Average amount (mm) |
| SAMIR-Reservoir | 0.37 | 595 | 0.69 | 693 | 0.34 | 621 | 0.62 | 719 | 509 | 587 |
| SAMIR-SIDRA | 0.32 | 590 | 0.65 | 694 | 0.38 | 617 | 0.66 | 720 | | |

---

## Author Comment (AC2)

Figure 1 with index:

[Figure]

Table 4 with irrigation over the entire AB district:

| | Irrigation obtained for AB1 calibration | | | | Irrigation obtained for AB2 calibration | | | | In situ irrigation (entire AB district) | |
|---|---|---|---|---|---|---|---|---|---|---|
| | 2021 | | 2022 | | 2021 | | 2022 | | 2021 | 2022 |
| | RMSD (mm d$^{-1}$) | Average amount (mm) | RMSD (mm d$^{-1}$) | Average amount (mm) | RMSD (mm d$^{-1}$) | Average amount (mm) | RMSD (mm d$^{-1}$) | Average amount (mm) | Average amount (mm) | Average amount (mm) |
| SAMIR-Reservoir | 0.37 | 595 | 0.69 | 693 | 0.34 | 621 | 0.62 | 719 | 509 | 587 |
| SAMIR-SIDRA | 0.32 | 590 | 0.65 | 694 | 0.38 | 617 | 0.66 | 720 | | |

Table 4. RMSD between daily simulated and observed irrigation, seasonal (between May and October for each year) cumulated simulated irrigation obtained with site-calibrated SAMIR-Reservoir and site-calibrated SAMIR-SIDRA separately, and seasonal in situ irrigation averaged over the whole AB district (including non-irrigated plots).

---

## Author Response (AR1)

Dear Editor,

Thank you for your detailed response which helped us to improve the relevance and clarity of the article. We addressed all the remarks and questions and proposed additions and changes in the paper. Your comments are listed below in bold, followed by our answers and the quotations we propose to add to the manuscript.

Yours sincerely,

The authors

**Dear authors,**

**I agree with your assessment that the reviewers provided constructive feedback. I studeid your replies, and by and large you seem to have a good idea about how to revise yopur paper. I have some doubts about a few of your replies though, and I explain those below. I therefore request you to have a closer look at those to see if they merit a somewhat different reponse.**

**Yours sincerely,**

**Gerrit de Rooij**
**Editor**

**Reviewer 1:**

**The comments about calibration vs validation worry me a little. Successful calibration alone is not a measure of model performance or quality.**

Authors' response:

We fully agree that a successful calibration is not in itself a measure of a model's performance or quality. Rather, we believe it is a measure of the adequacy between the complexity (number of parameters or dimensionality) of the model and the data available to calibrate it. This question is of great importance for drainage modeling, due to the critical lack of in situ drainage measurements (and more generally the lack

of knowledge of subsurface transfers) in the real world. A model can be properly calibrated and validated locally on a monitored basin, but extensive literature shows that the results are difficult to apply elsewhere in the absence of data. This study therefore examines the calibration of 4 different models with varying degrees of complexity, using 2 data sources (in situ data and default parameters/literature).

**Your use of default/literature parameters is a sort of validation, but still: what is the point of calibrating a model if it has to be recalibrated for every season?**

Authors' response:

We fully agree that variable parameters and the need to recalibrate models for every season are undesirable. However, we believe that the variability of model parameters, when it occurs, can be a powerful indicator of a key process missing in modeling. We argue that interpreting these variabilities is a means of improving current models and possibly their calibration possibilities. As an example, we demonstrate in this study that the variability of the lumped RU model parameters is mainly due to the non-linearity of the relationship between irrigation and deep percolation, poorly represented at the sub-basin scale (but correctly represented by the SAMIR model spatialized at the irrigated plot scale). The variation of model parameters also indicates a lack of or poor representation of physical processes (e.g., lateral subsurface flows, root growth, evapotranspiration), suggesting the need to better simulate them.

We propose to clarify the objectives of the accuracy assessment by modifying the lines 111-115:

*"Precision evaluation aims to investigate the models' strengths and weaknesses by calibrating and validating them over the same period. The accuracy evaluation aims to determine i) whether it is possible to estimate drainage when no in situ drainage data is available for calibration (which is the case for most irrigation districts), i.e., under non-optimal calibration conditions, and ii) which of the models evaluated performs best under these conditions."*

**You argue this is due to the semi-empirical nature of the models, but is this not beside the point? If a model is successful at calibration but fails at validation, it can only fit a curve well. In that case, the parameters have high descriptive value, but low predictive value. This may be the fundamental reason for the good performance of the simplest model: by sacrificing descriptive value, the model may have gained predictive value because it models the key processes with some accuracy.**

We agree with the reviewer that the main purpose of developing advanced drainage models is to improve model predictions. Frankly, we don't know exactly why the simplest model performs better than the more complex ones. However, we think this is an important result as it indicates that the descriptive complexity of a model may not be useful for predictive purposes. One example is the difficulty of relating drainage model parameters to physically measurable soil properties, although semi-empirical models can represent drainage satisfactorily (Golmohammadi et al., 2020; Henine et al., 2022; Jeantet al., 2021).

We propose to emphasize that our results revealed that the descriptive complexity of the models may not be useful for predictive purposes, by adding the following sentence to lines 597-600 in Section 3.2.3.

*"The fact that SAMIR-Reservoir shows higher accuracy than the more complex models using SIDRA is an interesting result since it shows that the descriptive complexity of the models may not be useful for predictive purposes. One reason for this is the difficulty of linking SIDRA soil parameters to physically measurable soil properties."*

**I think you need to better explain how models that have fluctuating parameter values can be used in practice. It might affect the usefulness of the models as decision support tools, for instance. This needs to be addressed thoughtfully.**

Authors' response:

Fluctuating parameter values are undesirable. However, when it does occur, it is a fact that deserves the attention of researchers, as it can be a powerful indicator of a key process missing from the modeling and/or a mismatch between the complexity of the model and the data actually available to calibrate it correctly. Models need to be more robust. To do so, it is necessary to identify poorly simulated or unsimulated processes and consider them in the models.

We propose to emphasize this point by adding this sentence to the lines 333-335 in Section 2.3 "Strategy for evaluating the models' precision":

*"Analysis of the variability of the values of the calibrated parameters between the two periods provides information on the predictive capacity of the models. If the values are close from one period to another, this suggests that the model robustness is high. If they are not, this indicates a low level of robustness."*

We also propose to underline the consequence and potential causes of these fluctuating calibrated parameter values by adding the following subsection to Section 3:

*"3.1.5 Variability of calibrated parameter values between the two periods analyzed*

*We can see from Table 6 that the calibrated values of most parameters vary between the two periods for a given sub-basin. These variations indicate a lack of predictive capacity of the models, at least between the two periods analyzed (using the parameters values obtained from calibration on the first period, the models fail to predict the second period). We believe this is due to the semi-empirical nature of the models. Indeed, parameter values vary between the two periods to compensate for the fact that physical processes (e.g., lateral subsurface flows, root growth, evapotranspiration), which vary between the two periods, are either too empirically simulated or neglected."*

In addition, we propose emphasizing the consequences of parameters value variability on the usefulness of decision support tools and on the need to increase model robustness by adding the following paragraph to Section 3.1.6 Recommendations and perspectives.

*"To support the strength of these recommendations and to ensure that the models can be ultimately used as decision support tools with confidence, we believe it is necessary to improve the robustness of the models by better simulating certain physical processes (e.g., lateral subsurface flows, root growth, evapotranspiration). Indeed, the variability of calibrated parameter values between the two periods indicates their limited robustness."*

As well as this paragraph in section 3.2.4 Recommendations and perspectives.

*"SAMIR-Reservoir offers a certain consistency with the observed data and could provide an approximate idea of the drainage dynamics and amount occurring in an ungauged irrigated sub-catchment. Figures 6.e, 6.f, 7.e, and 7.f show that SAMIR-Reservoir 2000 simulations are less dispersed than for the other three models, suggesting greater robustness. For decision support, we therefore recommend the use of SAMIR-Reservoir. However, its accuracy should be further evaluated on other sites. Furthermore, its accuracy could be improved i) by modeling lacking physical processes and ii) by investigating a link between the depletion coefficient parameter $\omega$ and the soil texture or the characteristics of the drainage network."*

Finally, we propose to emphasize the variability of calibrated parameter values and the need to improve model robustness in the conclusion, by adding this sentence to lines 644-648:

*"In addition, the calibrated parameter values vary between the two periods analyzed for a given sub-basin. This indicates that the models have limited predictive capacities (robustness). It is therefore necessary to identify the processes poorly simulated or not simulated at all, such as lateral subsurface flows, root growth, evapotranspiration, and to better take them into account in the models."*

**In your response to the comment by the reviewer about the vast difference in size between two irrigation districts, you suggest an edit to clarify the differences between the two that does not mention the size.**

Authors' response:

Our edit aimed to highlight the differences in terms of irrigation and drainage between the two sub-basins. We had already mentioned the difference in size line 162: "They collect drainage water from areas of 116 and 2050 hectares each, forming two sub-basins, AB1 and AB2 (see Fig. 2)". The difference in crop type between the two sub-basin is also shown in Table 1.

**Reviewer 2:**

**Your proposed edit to the comment about saline groundwater is incomplete. You are correct that irrigated fields with relatively shallow groundwater tables need drainage to allow leaching of salts from the root zone. But the reviewer alluded to the need to lower the groundwater table when the groundwater is saline to such a level that capillary rise into the root zone is no longer possible (salinization from below).**

Authors' response:

Thank you for this comment. Indeed, the distinction between salinization from above and salinization from below needs to be clarified in the revised manuscript. We propose editing the paragraph at lines 39-43 with the following sentences:

*"Drainage systems are generally installed to prevent waterlogging during heavy rainfall (through a sudden rise in the water table), to facilitate salt leaching (particularly when the irrigation water has high salt concentrations), and/or to maintain a low water table to avoid salt accumulation in the root zone by capillary rise (particularly when the groundwater has high salt concentrations)."*

**The reply to comment about the groundwater table only being a function of time includes a suggested edit that does not address this. The new text does not explain that 'groundwater table' refers to the phreatic level at the midpoint between drains. The accompanying explanation makes this clear, but his information belongs in the main text.**

Authors' response:

Agreed. We propose to modify line 285 as follows:

*"where h is the water table at the midpoint between drains (m), K the horizontal hydraulic conductivity (m d-1), μ the drainable porosity (m3 m-3), and C a water table shape factor (-) equal to 0.904."*

**In your response in the comment on the Monte Carlo-type simulation about the reviewers' confusion about the way L was treated, you use the term' mid-drain spacing' to define L. This will add to the confusion. Either it is the 'drain spacing', or simply 'half the drain spacing'. If you state that 6 m is a value for L frequently found in the literature, you may want to back that up by a few references.**

Authors' response:

Thank you for this comment. It is true that the term 'mid-drain spacing' lacks clarity. We have replaced 'mid-drain spacing' with 'half the drain spacing' wherever the term was used in the manuscript. We have also added the reference Jeantet al. (2021) to justify the statement that a value of 6 m for L is frequently found in the literature.

**In your reply to the comment about Fig 6 and 7 you state that the calibrated Sinter values are much lower than those reported in the literature. The edit you suggest indicates that this is the result of spatially heterogeneous irrigation. Does this not imply that RU is of limited use for irrigation districts that are not under monoculture? If so, this should be discussed.**

Authors' response:

Indeed, calibrated Sinter values are low because RU does not use spatially heterogeneous irrigation as forcing (it simulates an average plot). This implies that RU's ability to predict drainage is limited to areas where agricultural and irrigation practices are homogeneous, such as sites under monoculture.

We propose to emphasize this point in lines 576-578:

*"This implies that without calibration with drainage data, models based on RU are only effective in contexts where irrigation practices are homogeneous, e.g., on sites under monoculture."*

**References**

Golmohammadi, G., Rudra, R. P., Parkin, G. W., Kulasekera, P. B., Macrae, M., and Goel, P. K.: Assessment of Impacts of Climate Change on Tile Discharge and Nitrogen Yield Using the DRAINMOD Model, Hydrology, 8, 1, https://doi.org/10.3390/hydrology8010001, 2020.

Henine, H., Jeantet, A., Chaumont, C., Chelil, S., Lauvernet, C., and Tournebize, J.: Coupling of a subsurface drainage model with a soil reservoir model to simulate drainage discharge and drain flow start, Agricultural Water Management, 262, 107318, https://doi.org/10.1016/j.agwat.2021.107318, 2022.

Jeantet, A., Henine, H., Chaumont, C., Collet, L., Thirel, G., and Tournebize, J.: Robustness of a parsimonious subsurface drainage model at the French national scale, Hydrology and Earth System Sciences, 25, 5447–5471, https://doi.org/10.5194/hess-25-5447-2021, 2021.